# In Vitro Anti-Orthohantavirus Activity of the High-and Low-Molecular-Weight Fractions of Fucoidan from the Brown Alga *Fucus evanescens*

**DOI:** 10.3390/md19100577

**Published:** 2021-10-15

**Authors:** Natalia V. Krylova, Artem S. Silchenko, Anastasia B. Pott, Svetlana P. Ermakova, Olga V. Iunikhina, Anton B. Rasin, Galina G. Kompanets, Galina N. Likhatskaya, Mikhail Y. Shchelkanov

**Affiliations:** 1G.P. Somov Institute of Epidemiology and Microbiology, Rospotrebnadzor, Selskaya Street, 1, 690087 Vladivostok, Russia; pott_a.b@mail.ru (A.B.P.); olga_iun@inbox.ru (O.V.I.); galkom1965@gmail.com (G.G.K.); adorob@mail.ru (M.Y.S.); 2G.B. Elyakov Pacific Institute of Bioorganic Chemistry, Far-Eastern Branch of the Russian Academy of Science, Prospect 100 let Vladivostoku, 159, 690022 Vladivostok, Russia; artem.silchencko@yandex.ru (A.S.S.); swetlana_e@mail.ru (S.P.E.); abrus__54@mail.ru (A.B.R.); galin56@mail.ru (G.N.L.)

**Keywords:** fucoidans, fucanase GH107, orthohantavirus, AMRV, antiviral activity, molecular docking

## Abstract

The Hantaan orthohantavirus (genovariant Amur–AMRV) is a rodent-borne zoonotic virus; it is the causative agent of haemorrhagic fever with renal syndrome in humans. The currently limited therapeutic options require the development of effective anti-orthohantavirus drugs. The ability of native fucoidan from *Fucus evanescens* (FeF) and its enzymatically prepared high-molecular-weight (FeHMP) and low-molecular-weight (FeLMP) fractions to inhibit different stages of AMRV infection in Vero cells was studied. The structures of derivatives obtained were determined using nuclear magnetic resonance (NMR) spectroscopy. We found that fucoidan and its derivatives exhibited significant antiviral activity by affecting the early stages of the AMRV lifecycle, notably virus attachment and penetration. The FeHMP and FeLMP fractions showed the highest anti-adsorption activity by inhibiting AMRV focus formation, with a selective index (SI) > 110; FeF had an SI of ~70. The FeLMP fraction showed a greater virucidal effect compared with FeF and the FeHMP fraction. It was shown by molecular docking that 2*O*-sulphated fucotetrasaccharide, a main component of the FeLMP fraction, is able to bind with the AMRV envelope glycoproteins Gn/Gc and with integrin β3 to prevent virus–cell interactions. The relatively small size of these sites of interactions explains the higher anti-AMRV activity of the FeLMP fraction.

## 1. Introduction

Orthohantaviruses (family *Hantaviridae*, order *Orthohantavirus*) are enveloped, negative-stranded RNA viruses that belong to the most widely distributed zoonotic rodent-borne viruses [1,2]. These viruses are the etiological agents of two clinical forms of human disease: haemorrhagic fever with renal syndrome (HFRS) in Eurasia and hantavirus cardio-pulmonary syndrome in the Americas. There are between 150,000 and 200,000 cases of orthohantavirus disease worldwide each year, with the majority of HFRS cases occurring in Asia, mostly in China [3,4,5]. In the Russian Federation, HFRS takes a leading place among all natural focal human diseases [6]. In the Russian Far East, the annual HFRS morbidity is caused by two orthohantaviruses: Hantaan (genovariants Far East and Amur) and Seoul [7,8].

So far, the only licensed antiviral drug well described for orthohantaviruses is ribavirin, a broad-spectrum inhibitor of RNA virus replication [4,9]. Despite the beneficial effects of ribavirin therapy in the early phases of HFRS, resulting in a decreased risk of developing haemorrhagic syndrome and severe renal failure, the non-negligible side effects of this drug limit its use [10,11]. Thus, the limited range of antiviral drugs and the lack of licensed vaccines against orthohantavirus infection that could be widely used necessitate the search and development of new antivirals for HFRS treatment.

The recent decades have been characterised by active research in the field of drug development on the basis of polysaccharide macromolecules from marine hydrobionts. Among these are the sulphated fucose-containing polysaccharides from brown algae–fucoidans [12]. Increased interest in these polysaccharides is related to their low toxicity and wide spectrum of biological activity: antioxidant, anti-inflammatory, antitumor, antiviral, etc. [13,14,15]. The structure of their main chain, molecular weight, content and locations of sulphate and acetate groups are crucial for the biological properties of fucoidans [16,17,18,19]. Even though there has been obvious progress in the development of fucoidan-based drugs, there are currently no registered medicines because obtaining standardised batches of fucoidans is difficult and the information about the relationship between the structural and biological characteristics of fucoidans is limited.

It has been shown that fucanases are effective tools for studying the structures of fucoidans and for modification of native fucoidans, allowing the acquisition of derivatives that are easier to characterise [20,21]. Fucanases (syn. fucoidanases or fucoidan-hydrolases) are enzymes that catalyse the hydrolytic cleavage of α-glycosidic bonds between sulphated L-fucose residues within the polymer chains of fucoidans. These enzymes belong to the 107 or 168 families of glycoside hydrolases according to the Carbohydrate-Active enZymes (CAZy) database [22,23]. The study of the biological activity of the fucoidan fractions with defined structures is one of the important stages on the way to the development the fucoidan-based antivirals. 

Thus, the aim of present work was to compare the in vitro anti-orthohantavirus activity of the fucoidan from the brown alga Fucus evanescens (FeF) and its enzymatically prepared derivatives, which have distinct structures.

## 2. Results

### 2.1. Preparation of Enzymatically Modified Fucoidan Derivatives

Previously, we showed that fucanases of the GH107 family can be used to produce fucoidan derivatives with regular structures [20,21,24]. In the present work, we used the recombinant fucanase FFA2 of the marine bacterium *Formosa algae* KMM3553T (Figure 1A) to produce fragments from FeF isolated from brown algae Fucus evanescens. Enzymatically depolymerised products of FeF were separated into high-molecular-weight (FeHMP) and low-molecular-weight (FeLMP) fractions using 75% ethanol (Figure 1B). The product yields were 43% and 57% for the FeLMP and FeHMP fractions, respectively.

The structures of the FeHMP and FeLMP fractions were further analysed by nuclear magnetic resonance (NMR) spectroscopy. One-dimensional (1D) and two-dimensional (2D) NMR spectra of the FeHMP fraction coincide with those of a HMP fraction obtained previously [20]. This fraction had a regular structure with a repeating disaccharide unit [→3)-α-L-Fucp2,4S-(1→4)-α-L-Fucp2S(1→]. The average molecular weight of the FeHMP according to size-exclusion chromatography (SEC) analysis is about 71 kDa (Appendix A and Appendix A).

The FeLMP fraction was investigated by 1D NMR spectroscopy (Figure 2). Its 1H spectrum featured a large number of signals in the anomeric area. To assign the signals, the 1D total correlation spectroscopy (TOCSY) method was applied. There were two close signals at 5.48–5.49 ppm, corresponding to the reducing ends of fucooligosaccharides. The 1D TOSCY experiment showed their spins interact with the spins of the signals at 4.51, 4.04–4.06 and 4.08–4.09 ppm. Our previous research [25] showed such signals correspond with the reducing α-L-fucopyranose residue sulphated at C2. The structures of other residues were deduced by using the same method. The signal at 5.41 ppm correlated with the signals at 4.53, 4.16 and 4.11 ppm, indicating it to be the anomeric signal of a residue with the →3)-α-L-Fucp2S(1→ structure. A similar structure was deduced for a residue whose anomeric signal was 5.27 ppm. The difference in the anomeric shifts of these residues is presumably due to the influence of their neighbor residues. There were several close residues located at 5.34–5.35 ppm. The 1D TOCSY spectrum revealed one of them to belong to a residue with the →3)-α-L-Fucp2,4S(1→ structure, while the others include a non-reducing end with the α-L-Fucp2S(1→ structure and residues with the →4)-α-L-Fucp2S(1→ structure. Anomeric proton signals at 5.29–5.31 ppm also seem to belong to residues with the →3)-α-L-Fucp-2,4S(1→, α-L-Fucp2S(1→ and →4)-α-L-Fucp2S(1→ structures, as well as one or several residues with the →3)-α-L-Fucp-2S(1→ structure. The low intensity signal at 4.97 ppm in the 1H spectra attributed to H4 of the →3)-α-L-Fucp2,4S residue indicates the presence of a small amount of 4O-sulphation (Figure 2). Therefore, the oligosaccharides of the FeLMP fraction mainly comprise the repeating disaccharide unit [→3)-α-L-Fucp2S-(1→4)-α-L-Fucp2S(1→], with some oligosaccharides containing minor inclusions of sulphates at C4 of →3)-α-L-Fucp2S residues (Figure 1B).

The FeLMP fraction according to carbohydrate polyacrylamide gel electrophoresis (C-PAGE) analysis consists of a mixture of oligosaccharides with different degrees of polymerisation (DP) (Figure 3C). The DP of oligosaccharides in the FeLMP fraction was calculated by SEC using 2O-sulphated fucooligosaccharides as reference standards (Figure 3B). The FeLMP fraction contains oligosaccharides with a calculated DP from 4 to 16 with a dominant portion of oligosaccharides with a DP of 4 (Figure 3A).

### 2.2. Antiviral Activity of the Fucoidans against the Orthohantavirus Amur

We have previously shown that FeF and its FeHMP fraction inhibit the replication of some DNA and RNA viruses (HSV-1, HSV-2, ECHO-1 and HIV-1) [26]. In this work, we investigated the antiviral activity of FeF and its enzymatically prepared derivatives—FeHMP and FeLMP—against the RNA-containing virus orthohantavirus Amur (AMRV).

The cytotoxicity assessment of the tested fucoidans (FeF, the FeHMP fraction and the FeLMP fraction) and ribavirin as a reference compound were carried out by means of the methylthiazolyltetrazolium bromide (MTT) assay. The investigated compounds had low toxicity to Vero cells. The 50% cytotoxic concentrations (CC_50_) of all the fucoidans were above 2000 μg/mL, although ribavirin was more toxic to Vero cells, with a CC50 of 730 μg/mL (Appendix A). Further antiviral activity assays were performed using FeF, the FeHMP fraction and the FELMP fraction at a concentration below 500 μg/mL. 

The antiviral effect of the studied compounds on different stages of the orthohantavirus infection was studied by using the focus formation reduction assay. The following treatment schemes were used: pre-treatment of the virus (compounds were added directly to the virus suspension); pre-treatment of the cells (cells were treated with compounds for 1 h before infection); attachment (cells were co-treated with the virus and compounds at 4 °C); penetration (cells were infected with the virus at 4 °C and then treated with the compounds at 37 °C); and treatment of infected cells (compounds were added 1 h after infection). The results obtained from the focus formation reduction assay were used to calculate the concentration that inhibits 50% of the formation of the viral focus (IC_50_) and the selectivity index (SI) as the ratio of CC_50_ to IC_50_ for each of the compounds (Figure 4 and Appendix A).

Pre-treatment of AMRV with different concentrations (5–500 μg/mL) of the tested compounds (direct virucidal effect) showed a moderate antiviral activity of FeF and the FeHMP fraction (each had a mean IC_50_ of ~100 μg/mL and SI of ~19). The FeLMP fraction exhibited higher virucidal activity compared with the FeHMP fraction: the IC_50_ of the FeLMP fraction was 2.0 times lower and the SI of the FeLMP fraction was higher than the SI of FeF and the FeHMP fraction (*p* ≤ 0.05) (Figure 4).

The treatment of Vero cells with FeF or the FeHMP fraction before AMRV infection (preventive effect) revealed a more effective inhibition of virus replication (each had a mean IC_50_ of ~67 μg/mL and SI of ~30) than when they affected the virus directly. At the same time, the FeLMP fraction protected Vero cells against orthohantavirus infection, with an SI 1.5-times higher than the SI of FeF and the FeHMP fraction (*p* ≤ 0.05). 

The tested compounds exhibited the highest anti-AMRV activity at the stage of virus attachment to cells. The FeHMP and FeLMP fractions inhibited the binding of the virus to cells (each had a mean SI of ~110) significantly more compared with FeF (SI = 72) (*p* ≤ 0.05). There was no significant difference in the antiviral effect between the FeHMP and FeLMP fractions (*p* ˃ 0.05).

The penetration assay showed that the effect of the tested fucoidans on virus entry into cells and revealed a significant reduction in virus focus formation (each had a mean SI of ~67). However, the difference between FeF and its derivatives was insignificant (*p* ˃ 0.05).

The application of fucoidans after virus adsorption and penetration to cells (treatment of infected cells) had a weak effect on the AMRV replication (each had a mean SI of ~4.6) (Figure 4 and Appendix A). It should be noted that this method of compound application was the only one for which ribavirin showed quite high viral inhibitory activity (SI ˃ 35), which is consistent with the data from other publications [27,28,29].

Overall, the data obtained showed that both FeF and its enzymatically prepared derivatives, when applied during the early stages of AMRV infection, are able inhibited virus attachment and penetration effectively into Vero cells.

### 2.3. Computer Modelling

#### 2.3.1. Molecular Docking of 2O-Sulphated Fucooligosaccharide with the Integrin β3 Epitope

A promising strategy for the development of new treatments and to prevent virus infections is inhibiting virus attachment and entry to the cells. The anti-AMRV activity of the fucoidan derivatives obtained in this work was highest upon simultaneous treatment of Vero cells and virus particles (attachment test) with the FeLMP fraction (Figure 4). It can be assumed that the FeLMP fraction interacts with cellular receptors and affects the binding of receptors to viral envelope glycoproteins, reducing the attachment and penetration of viruses into the cells.

Integrin β3 is a cellular receptor for the binding and entry of orthohantaviruses into cells [30]. Antibodies that neutralise orthohantaviruses bind to integrin β3 [31]. Pre-treatment of cells with anti-integrin β3 antibodies such as c7E3 or its Fab fragment ReoPro prevents orthohantavirus entry. It was shown that a cyclic peptide interacting with the integrin β3 epitope for ReoPro inhibits the activity of orthohantavirus in the same way as a neutralising antibody [32]. We studied in silico the interaction of 2O-sulphated fuco-tetrasaccharide, the main structural component of the FeLMP fraction, with the integrin β3 epitope for ReoPro (Figure 5). Molecular docking revealed that the oligosaccharide interacts with the integrin β3 for ReoPro epitope (Figure 5). Binding of fucooligosaccharide to the integrin β3 epitope occurs through electrostatic interactions as well as ionic and hydrogen bonds. The results of molecular docking suggest that the putative mechanism of the antiviral action of 2O-sulphated fucooligosaccharide upon pre-treatment of Vero cells involves the binding of fucooligosaccharide to the same integrin β3 epitope as neutralising antibodies, a phenomenon that is necessary for interaction with orthohantaviruses.

#### 2.3.2. Modelling the 3D Structure of the AMRV Glycoprotein Ectodomains Gn and Gc and the Spike Complex Tetramer (Gn-Gc)_4_

Alignment of the amino acid sequence of the Gn glycoprotein ectodomain of AMRV (Uniprot ID A3FEU7) and the crystal structure of the Hantaan virus (HNTV) Gn glycoprotein (PDB ID 6y6p) showed an amino acid residue identity of 83.8%. The superposition of the model AMRV Gn and the template HNTV Gn showed that the root-mean-square deviation (RSMD) for Cα-atoms is 0.76 Å. Alignment of the amino acid sequence of the Gc glycoprotein of AMRV (Uniprot ID A3FEU7) and the crystal structure of the Andes orthohantavirus (ANDV) envelope glycoprotein Gc (PDB ID 6y5f) showed an amino acid residue identity of 64.4%. Theoretical models of the spatial structure of the AMRV Gn and Gc glycoprotein ectodomains were built by the program MOE 2020.0901 and were protonated three-dimensionally before the final energy minimisation (Figure 6). The superposition of the model AMRV Gc and the template ANDV Gc showed that the RSMD value for 417 Cα-atoms is 0.94 Å. High-precision models of the AMRV Gn and Gc glycoproteins were obtained and used to construct a spike heterotetramer (Figure 6) and molecular docking with the ligand 2O-sulphated fucooligosaccharide.

#### 2.3.3. Molecular Docking of Fucooligosaccharide into the Epitopes for Neutralising Antibodies at the AMRV Gn and Gc Ectodomains

Virus attachment and entry represent the first and most fundamental steps in virus infection. The attachment and cellular entry of orthohantaviruses are mediated by interaction of the viral glycoproteins Gn/Gc with cell receptors. The crystal structures of the complexes of Gn and Gc glycoproteins with cellular receptors have not been established. Orthohantavirus neutralising antibodies bind to epitopes at the spike glycoproteins Gn and Gc and influence interactions with cellular receptors. The structures of the epitopes of neutralising antibodies on the glycoproteins Gn and Gc of orthohantaviruses have been established [33]. Epitopes are located on the outer surface of Gn and Gc glycoproteins. The crystal structure of the Gn complex with a neutralising antibody showed that the epitope includes positively charged amino acid at residues 83–85 [34]. This epitope can potentially serve as a binding site for a 2O-sulphated tetrasaccharide. To test this assumption, molecular docking of 2O-sulphated tetrasaccharide into this site was carried out. The oligosaccharide binds to this Gn epitope through hydrogen and ionic bonds (Figure 7). It is possible that the binding of a fucooligosaccharide to this epitope prevents or hinders interaction with cellular receptors.

An epitope on the Gc glycoprotein for a neutralising monoclonal antibody (3D8) was established and the epitope was shown to contain a sequence of positively charged amino acids at residues 893–895 [35]. Molecular docking to this site on the pre-fusion structure of Gc showed that 2O-sulphated fucooligosaccharide interacts due to hydrogen-acceptor and ionic bonds.

Thus, molecular docking has shown that 2O-sulphated fucooligosaccharide can interact with the AMRV spike complex from the Gn–Gc heterotetramer at sites that overlap with epitopes for neutralising antibodies (Figure 7 and Figure 8). Binding of fucooligosaccharide to positively charged sites of Gn/Gc ectodomains of orthohantavirus may explain the mechanism of virucidal action of negatively charged fucooligosaccharides of the FeLMP fraction.

Overall, by using bioinformatics approaches, spatial models of AMRV glycoproteins Gn and Gc were constructed and molecular dockings with fucooligosaccharide were carried out. The putative fucooligosaccharide binding sites on AMRV spike glycoproteins, which overlap with epitopes of orthohantavirus neutralising antibodies, were determined. The putative mechanism of the virucidal action of fucooligosaccharide on orthohantaviruses may be associated with interaction with epitopes for neutralising antibodies of spike proteins important for interaction with cellular receptors.

## 3. Discussion

In recent years, it has been demonstrated that fucoidans from various brown algae exhibit antiviral activity both in vitro and in vivo against various DNA-containing viruses [13,36,37] and RNA-containing viruses [38,39,40]. Meanwhile, there are only a few reports about the anti-orthohantavirus activity of fucoidans from algae *Saccharina cichorioides* (formerly named *Laminaria cichorioides*), *Saccharina japonica* (formerly named *Laminaria japonica*) and *F. evanescens* [41,42,43,44]. Because there are difficulties in the standardisation of fucoidan extraction [45], mainly due to their heterogeneous structure [46], an important task is to identify the structural fucoidan fragments responsible for their biological activity, including antiviral activity. In present work, we investigated the anti-orthohantavirus activity of FeF and its enzymatically prepared FeLMP and FeHMP fractions, which have defined structures.

### 3.1. Structural Organisation of FeF

Structural characterisation of fucoidan samples is an important step when investigating the structure–bioactivity relationships of these biopolymers. The use of fucanases in tandem with NMR spectroscopy can provide important information about the structure of fucoidans [21,24]. It was previously shown that fucanase FFA2 cleaves α-1→4-glycosidic bonds within 2*O*-sulphated fucoidan regions and does not depolymerise fucoidan regions that contain other sulphation (Figure 1A) [25]. This feature made it possible to determine the main structural fragments that are present in FeF, as well as to estimate the distribution of sulphate groups along the main chain of this polysaccharide. Presumably, FeF has a complex structural organisation, with several regions containing different forms of sulphation (Figure 1B). This has been confirmed by the structure of the obtained products of enzymatic hydrolysis. FFA2 depolymerises the 2*O*-sulphated FeF regions, resulting in oligosaccharides (the FeLMP fraction) and accumulation of the resistant FeHMP fraction with a regular structure [→4)-Fucp2S-(1→3)-Fucp2,4S-(1→]. Accumulation of the FeHMP fraction with a molecular weight of about 71 kDa indicates the presence in FeF of extended regions with the regular [→4)Fucp2S-(1→3)-Fucp2,4S(1→] structure. This structure comprises about half of FeF. This finding is consistent with previously obtained data [24]. The FeLMP fraction contains oligosaccharides composed of the repeating α-1→3- and α-1→4-linked L-fucopyranose residues sulphated mainly at C2. Additional sulphates that occupy position C4 in 2O-sulphated L-fucopyranose residues of some oligosaccharides were also detected. Therefore, the remaining part of FeF is predominantly [→4)Fucp2S-(1→3)-Fucp2S(1→] with minor inclusions of 4O-sulphation. It can be speculated that minor inclusions of 4O-sulphation are a part of the transition zone between the regular 2O-sulphated and regular alternating 2O-sulphated and 2,4diO-sulphated fucoidan regions (Figure 1B). A similar structural organisation has been described for mammalian glycosaminoglycans, which contain regions with different sulphation [47,48].

### 3.2. Comparative Analysis of the Anti-AMRV Activity of FeF and Its Derivatives

To identify possible relationships between the structure and anti-orthohantavirus activity of fucoidans, we investigated the effect of FeF and its enzymatically prepared derivatives (the FeHMP and FeLMP fractions) at different stages of AMRV infection. FeF, the FeHMP fraction and the FeLMP fraction directly affected viral particles (virucidal effect), increased the resistance of Vero cells to infection (prophylactic effect) and inhibited the attachment and penetration of AMRV to cells (Figure 4 and Appendix A).

Comparative analysis of the anti-AMRV activity of the studied fucoidans showed that the FeLMP fraction had a higher virucidal activity (directly acting on the virus) and prophylactic effect (directly acting on the Vero cells) than FeF and the FeHMP fraction (*p* ≤ 0.05). The high virucidal and prophylactic activities of the FeLMP fraction relative to the other fucoidans tested may be a result of multivalent interactions of the FeLMP fraction with both the virus and its cellular receptors. The higher antiviral activity of the FeLMP fraction may be associated with its lower DP compared with FeF and the FeHMP fraction, and may also indicate the relatively small size of the target sites required to block virus–receptor interactions.

The tested fucoidans exerted the greatest inhibition on AMRV replication during the early stages of the viral life cycle—virus attachment and penetration—but had no effect on post-entry events. Both modified fucoidans (the FeHMP and FeLMP fractions) showed a significantly higher level of anti-adsorption activity (SI ˃ 110) compared with FeF (SI ~70) (*p* ≤ 0.05). In addition, all three tested fucoidans displayed a similarly high inhibitory effect (SI ~67) of blocking the penetration of AMRV into cells. We believe that the high anti-adsorption activity of the studied fucoidans against AMRV stems from the ability of sulphated polysaccharides to inhibit the attachment of viruses to target molecules on the cell surface [49,50].

### 3.3. The Proposed Mechanism of Action of Fucoidans and Sulphated Fucooligosaccharides on AMRV

As discussed above, FeF and its derivatives (the FeLMP and FeHMP fractions) affect the early stages of the viral life cycle, including reducing the ability of AMRV to attach to cells, increasing the resistance of Vero cells to the infection and inhibiting the attachment and penetration of AMRV to cells. These findings indicate that the main targets for the action of the fucoidans under study could be both viral envelope glycoproteins and cell surfaces receptors for virus entry into host cells. It is known that orthohantaviruses use their envelope glycoproteins Gn and Gc to attach to and penetrate host cells [51,52]. Several cellular proteins have been proposed as receptors for orthohantavirus entry, including integrins (β1–3), membrane proteins of the complement regulatory system (such as the decay-accelerating factor [DAF/CD55] and a receptor for the C1 component of the complement [gC1qR]) and a transmembrane protein of the cadherin superfamily, protocadherin-1 (PCDH1) [53,54]. One of the first integrins characterised as a cellular receptor for orthohantaviruses was integrin αVβ3, antibodies to which effectively blocked infection in various cells, including Vero, caused by virulent and avirulent orthohantaviruses [30,55,56]. It has recently been demonstrated that some integrins can also be receptors for fucoidan [50,57]. Indeed, fucoidan isolated from *Sargassum fusiforme*, by binding to integrin αVβ3 on liver cancer cells, triggers signaling pathways by which it exerts anti-metastatic activity of these cancer cells [58].

Molecular docking showed that 2O-sulphated tetrasaccharide, a main component of the FeLMP fraction, can bind to the integrin β3 antibody epitope, neutralising the interaction with the envelope glycoproteins Gn/Gc of orthohantavirus. Moreover, this sulphated tetrasaccharide can interact with the AMRV envelope Gn/Gc heterotetramer at sites that overlap with epitopes for neutralising antibodies (Figure 7 and Figure 8). The data obtained indicate a putative mechanism of the anti-orthohantavirus action of fucoidans and oligosaccharides, in which they are able to block both cell receptors and viral proteins, preventing the virus from binding to and further penetrating the host cells. This mechanism has been confirmed by the results of an experiment in which a mixture of the AMRV particles and fucoidans added to the Vero cells leads to the highest anti-orthohantavirus effect (Figure 4). The binding of integrin β3 and Gn/Gc ectodomains to the tetrasaccharide occurred mainly between the positively charged sites of proteins and the negatively charged 2O-sulphate groups of the oligosaccharide, which indicates the importance of sulphation in this process. These sites of interaction between integrin β3 and the orthohantavirus Gn/Gc ectodomains are relatively small; hence, they do not require a high DP of fucoidans to block them. This is consistent with experimental evidence in which the IC50 for the FeLMP fraction is lower compared with FeF and the FeHMP fraction (Figure 4). Thus, the sulphated fucooligosaccharides can be promising anti-orthohantavirus antivirals.

It should be noted that the rather high activity of fucoidans in the experiment examining the AMRV penetration of cells, in which the virus had already bound the cells, cannot be explained by simple blocking of the cell receptors and orthohantavirus envelope proteins by fucoidans. There are probably alternative mechanisms of fucoidan action on the orthohantavirus penetration that are still poorly understood. FeF and its derivatives may inhibit infectivity by blocking one or more post-entry steps of virus replication. A similar suggestion was made when studying the action of fucoidan from *Cladosiphon okamuranus* on Newcastle disease virus (NDV) [59]. Thus, further study of the mechanisms of inhibition of post-entry orthohantavirus events by fucoidans is needed.

## 4. Materials and Methods

### 4.1. Virus and Cell Culture

In this study, the genovariant AMRV, a pathogenic in humans’ strain of the orthohantavirus Hantaan, was obtained from the collection of the Laboratory of Experimental Virology of the G.P. Somov Institute of Epidemiology and Microbiology, Rospotrebnadzor. AMRV was isolated from the lungs of the East Asian mouse *Apodemus peninsulae* in 1999 (GenBank number: AB071185.1) [60].

AMRV was grown in African green monkey kidney (Vero) cells using Dulbecco’s Modified Eagle’s Medium (DMEM, Biolot, St. Petersburg, Russia) supplemented with 7% fetal bovine serum (FBS, Biolot, St. Petersburg, Russia) and 100 U/mL of gentamycin (Dalkhimpharm, Khabarovsk, Russia) at 37 °C in a CO_2_ incubator. In the maintenance medium, the FBS concentration was decreased to 1%.

### 4.2. Studied Compounds

The samples of the brown alga *F. evanescens* were collected from Troitsa Bay, Sea of Japan, Far East of Russia, in July 2018. A crude fucoidan from F. evanescens was obtained as described previously [61]. Fucoidan fractions were isolated according to a reported method [62]. The recombinant fucanase FFA2 of marine bacterium *F. algae* KMM3553T was prepared according to a published protocol [20]. 2O-Sulphated fucooligosaccharides were prepared according to a previously published method [63] using fucanase FWf2 of the marine bacterium *Wenyingzhuangia fucanilytica* CZ1127T.

### 4.3. Preparation of Enzymatic Hydrolysis Products of Fucoidan

The products of enzymatic hydrolysis were obtained by the method described previously [20]. In brief, FeF was treated with fucanase FFA2 for 72 h at 34 °C. The hydrolysis products of FeF were deproteinised by heating in a water bath at 80 °C for 10 min and then centrifuged at 14,000× *g* for 40 min at 4 °C. Next, ice cold 96% ethanol was added to the fucoidan hydrolysis products to achieve a final concentration of 75%. The mixture was incubated in a freezer for 1 h and then centrifuged at 10,000× *g* for 20 min at 4 °C. The pellet contained high-molecular weight products, named the FeHMP fraction, and the supernatant contained low-molecular weight products, named the FeLMP fraction. The FeHMP fraction was dissolved in deionised water and freeze dried. The FeLMP fraction was dried on a rotary evaporator, dissolved in deionised water, desalted on a Superdex G-10 column and freeze dried.

For cytotoxicity and anti-AMRV activity determination, the tested compounds—FeF, the FeHMP fraction and the FeLMP fraction—were diluted in DMEM. A stock solution (10 mg/mL) of ribavirin (Sigma-Aldrich, St. Louis, MO, USA) was used as reference compound; it was dissolved in dimethyl sulfoxide (DMSO, Sigma-Aldrich, St. Louis, MO, USA) and stored at −20 °C. For experiments, it was diluted with DMEM to a final concentration of 0.5% DMSO.

### 4.4. Analytical Methods

The fucoidan samples (1 mg) were hydrolysed with 2 mL of 2 N trifluoroacetic acid (TFA) at 100 °C for 6 h, dried using a rotary evaporator, dissolved in 1 mL of deionised water, centrifuged and used for analysis. The content of sulphate groups in fucoidan fractions was determined by using the BaCl2-gelatine method [64,65]. The monosaccharide composition of fucoidan were determined by using a high-performance anion-exchange chromatography with pulsed amperometric detection (HPAEC-PAD) on an Agilent 1260 Infinity II (Agilent Technologies, Santa Clara, CA, USA) system with electrochemical detector DECADE Elite (Antec Scientific, Zoeterwoude, the Netherlands). The analysis was performed using a CarboPac PA1 column (25 cm × 4 cm, Thermo Fisher, Sunnyvale, CA, USA) with a linear gradient from 3 to 5 mM of NaOH for 26 min with a subsequent linear gradient from 100 to 200 mM of NaOAc in 0.1 M NaOH for 15 min.

#### 4.4.1. Determination of the DP of Oligosaccharides

The DP of the FeLMP fraction was determined by SEC, using an Agilent 1100 Series HPLC instrument (Agilent, Germany) equipped with a refractive index detector and a Superdex-30 column (15 mm × 500 mm) with. Sulphated oligosaccharides were eluted with 0.2 M NH4HCO3 aqueous solution at a flow rate 0.4 mL/min. The DP of sulphated fucooligosaccharides of the FeLMP fraction were estimated using 2O-sulphated tetra-, hexa-, octa- and deca-fucooligosaccharides as standards.

#### 4.4.2. Molecular Weight Determination of the Fucoidan Samples

Molecular weights of the fucoidan samples were determined by SEC, using an Agilent 1100 Series HPLC instrument equipped with a refractive index detector and series-connected SEC columns, Shodex OHpak SB-805 HQ and SB-803 HQ (Showa Denko, Japan). Elution was performed with aqueous 0.15 M NaCl at 40 °C, with a flow rate of 0.4 mL/min. The molecular weights of FeF, FeHMP and FeLMP were estimated using dextrans of 5, 10, 25, 50, 80, 250, 410 and 670 kDa (Sigma-Aldrich, Darmstadt, Germany) as reference standards.

#### 4.4.3. PAGE of Sulphated Oligosaccharides

The diversity of sulphated oligosaccharides presented in the FeLMP fraction was assessed by PAGE according to the method described previously [25].

#### 4.4.4. NMR Spectroscopy

NMR spectra were recorded using an Avance II-500 HD NMR spectrometer (Bruker, Hamburg, Germany). 1H and 1D TOCSY spectra were recorded for carbohydrate samples solutions in D2O at 35–40 °C, with acetone as the internal standard. The concentration of the samples was 5–15 mg/mL.

### 4.5. Virological Methods

#### 4.5.1. Focus Formation Assay

The focus formation assay was used to determine the AMRV titters [66]. Monolayers of Vero cells grown on 24-well plates (1 × 10^5^ cells/well) were infected with 10-fold serial dilutions of the virus (100 μL/well of each dilution) and incubated for 1 h at 37 °C. This was done in duplicate for each dilution, and a mock well was included that containing DMEM without virus. Then, the virus inoculum was removed, and each well was covered with 1.5 mL of 0.6% (w/v) carboxymethyl cellulose (CMC, ICN Biomedicals Inc., Aurora, OH, USA) in maintenance medium. The plates were then incubated at 37 °C (5% CO_2_) for 7–9 days, as the orthohantavirus replication cycle is known to be slower than most other RNA viruses, so harvesting orthohantaviruses usually take at least 7–10 days [67]. After incubation, the overlay was removed and the cells were washed with phosphate-buffered saline (PBS, pH 7.2). Next, cells were fixed with 96% cold ethanol for 15 min at room temperature, then the cells were washed twice with PBS and covered with anti-orthohantavirus antibodies (AMRV infection convalescent serum at a 1:32 working dilution). After incubation at 37 °C for 1 h, the cells were washed three times with PBS and covered with peroxidase-conjugated protein A (Sigma-Aldrich, St. Louis, MO, USA) at 37 °C for 1 h. Then, the cells were washed and a peroxidase indicator containing 0.05% diaminobenzidine peroxidase solution (DAB, Sigma-Aldrich, St. Louis, MO, USA) with metals (Metal Enhanced DAB Substrate Kit, Thermo Scientific, Waltham, MA, USA) was applied for 30 min as per the manufacturer’s instructions. The plates were dried, and the number of infected cell colonies (focus-forming units [FFU]) was calculated visually. The viral titer is expressed as FFU/mL. The titre of the AMRV was 104 FFU/mL; in further experiments the cells were infected with AMRV at a multiplicity of infection (MOI) of 0.01 FFU/cell.

#### 4.5.2. Cytotoxicity of the Fucoidans

The cytotoxicity of the studied fucoidans on Vero cells was evaluated using the MTT assay (Sigma-Aldrich, St. Louis, MO, USA) [67]. Briefly, confluent Vero cells (1 × 10^4^ cells/well) in 96-well microplates were incubated with various concentrations of compounds (0.2–2000 µg/mL) at 37 °C (5% CO_2_) for 7–9 days; untreated cells were used as controls. Then, MTT solution was added to cells at a concentration of 5 mg/mL and was incubated for 2 h at 37 °C. After the MTT solution was removed, formazan crystals were dissolved by the addition of isopropanol. The absorbance of the dissolved formazan was measured spectrophotometrically at 540 nm, with absorbance at 690 nm measured as background. Cytotoxicity is expressed as the CC_50_ of each tested compound, the concentration that reduced the cell viability by 50% compared with untreated cells. Experiments were performed in triplicate and repeated three times.

#### 4.5.3. Anti-AMRV Activity of the Fucoidans

The anti-orthohantavirus activities of the tested compounds were evaluated by measuring the reduction in the number of viral foci. The monolayer of Vero cells grown on 24-well plates (1 × 10^5^ cells/well) was infected with AMRV at a MOI of 0.01. Several fucoidan application schemes were investigated; each was performed in three independent replicates, with different concentrations of compounds (5–500 µg/mL). The plates were kept at 37 °C in a CO_2_ incubator for 7–9 days. These schemes were as follows:-Pre-treatment of virus with compounds. The virus was mixed with compounds at a 1:1 (*v*/*v*) ratio, pre-incubated for 1 h at 37 °C. Then the mixture was applied to the cellular monolayer and incubated for 1 h at 37 °C. After removing the mixture, the cells were washed with PBS and incubated with DMEM containing 0.6% carboxymethyl cellulose (CMC).-Pre-treatment of cells with compounds. A monolayer of cells was pre-treated with compounds for 1 h at 37 °C before infection. The cells were then rinsed with PBS to remove the compounds and infected with the virus for 1 h at 37 °C. Thereafter, the cells were washed with PBS to remove the unbound virus and incubated with maintenance DMEM with 0.6% CMC.-The attachment assay. The monolayer of cells was pre-chilled at 4 °C for 1 h and then treated with a mixture of virus and compound (1:1). After further incubation at 4 °C for another 3 h, the compounds and unabsorbed virus were removed by washing with cold PBS and the cells were incubated with DMEM with 0.6% CMC.-The penetration assay. The monolayer of cells, pre-chilled at 4 °C for 1 h, was infected with the virus and incubated at 4 °C another for 3 h. The unbound virus was removed with cold PBS and the infected cells were treated with the compounds and incubated for 1 h at 37 °C to allow viral penetration. Then, the unpenetrated virus was inactivated with citrate buffer (pH 3.0) and the cells were washed with PBS and incubated with DMEM with 0.6% CMC.-Treatment of infected cells. The monolayer of cells was infected with the virus at 37 °C for 1 h, then washed and overlaid with DMEM with 0.6% CMC containing different concentrations of the studied compounds.

In all assays, after 7–9 days of incubation, the viral foci were counted as described in Section 4.5.1. The antiviral effect of fucoidans was determined by the difference in the number of viral foci between the treated infected cells and untreated infected cells and is expressed as the percentage of viral foci reduction (RF, %). RF was calculated as follows: RF (%) = (1 − Ft/Fc) × 100, where Ft and Fc refer to the number of viral foci in the treated cells and the control cells (without compound), respectively. The IC_50_ of each compound was determined as the compound concentration that inhibited 50% of viral focus formation compared to the control. The SI was calculated as the ratio of CC_50_ to IC_50_ for each compound.

### 4.6. Computer Modelling

#### 4.6.1. Modelling the 3D Structure of the 2O-Sulphated Fucooligosaccharide

The 3D structure of the 2O-sulphated tetramer of fucooligosaccharide, the main component of the FeLMP fraction that exhibits antiviral activity against AMRV, was obtained using the module Build Carbohydrate of the program MOE 2020.0901 (Chemical Computing Group ULC. Molecular Operating Environment (MOE), 2020.09; Chemical Computing Group ULC: 1010 Sherbrook St. West, Suite #910, Montreal, QC, Canada, H3A 2R7, 2020. The oligosaccharide structure was optimised by solvation in water and energy minimisation with the Amber12:EHT forcefield using the MOE program.

#### 4.6.2. Homology Modelling of the AMRV Glycoproteins Gn and Gc

Homology models of Gn and Gc glycoproteins forming the spike complex of AMRV (Uniprot ID A3FEU7) were obtained by comparative modelling using the MOE homology module with the Amber12:EHT forcefield and default program parameters. The crystal structures of HNTV envelope glycoprotein Gn (PDB ID 6y6p) [68] and Andes orthohantavirus (ANDV) envelope glycoprotein Gc (PDB ID 6y5f) [68] with 83.8% and 64.4% identity, respectively, were used as templates. The AMRV spike tetramer model was built using the atomic model of ANDV glycoprotein spike tetramer (PDB ID 6zjm) [68].

#### 4.6.3. Molecular Docking of the AMRV Gn and Gc Glycoproteins and Human Integrin β3 with 2O-Sulphated Fucooligosaccharide

Molecular docking of the 2O-sulphated fucooligosaccharide with the AMRV Gn and Gc glycoproteins and with human integrin β3 (PDB ID 1u8c) [69] was performed using the Dock module of the MOE program. The structures of 30 complexes with score London dG and 5 complexes with score GBVI/WSA dG were calculated. The analysis of contacts in the complexes of the 2O-sulphated fucooligosaccharide with proteins was carried out using the Ligand Interaction module of the MOE program.

### 4.7. Statistical Analysis

Statistical processing of the data was performed using the Statistica 10.0 software (StatSoftInc, Tulsa, OK, USA). CC50 and IC50 were calculated by regression analysis of the dose–response curves. The results are presented as the mean ± standard deviation (SD). The differences between the parameters of the control and experimental groups were estimated using the Wilcoxon test. Differences were considered significant at *p* ≤ 0.05.

## 5. Conclusions

In this work, we have demonstrated the prospects for the use of enzymes in tandem with NMR spectroscopy to establish the structural features of fucoidans. The study of the structure of the FeLMP and FeHMP fractions of FeF, from the brown alga *F. evanescens*, showed that FeF contains areas with different sulphation. The structural organisation of FeF includes extended regular fragments, [→4)Fucp2S-(1→3)-Fucp2,4S(1→] and [→4)Fucp2S-(1→3)-Fucp2S(1→], as well as minor 4O-sulphated inclusions at →3)-Fucp2S residues. Presumably, minor 4O-sulphation is part of a transition zone between regions with regular 2O-sulphation and regular alternating 2O-sulphation and 2,4diO-sulphation.

Obtaining standardised fucoidan preparations with defined structures can be the basis of a strategy to develop promising anti-orthohantavirus agents. We have demonstrated for the first time the anti-orthohantavirus activity of FeF and its enzymatically prepared derivatives, the FeHMP and FeLMP fractions, which have defined structures. We showed that FeF and the FeHMP and FeLMP fractions exhibit significant anti-AMRV activity, mainly due to their ability to block the AMRV envelop glycoproteins Gc and Gn and integrin β3 of the host cells. The sites of interaction of between the glycoprotein Gn/Gc and integrin β3 with the fucoidans are relatively small; hence, the FeLMP fraction has relatively higher antiviral activity that FeF and the FeHMP fraction. The data obtained indicate that 2O-sulphated fucooligosaccharides are promising anti-orthohantavirus agents.

## Figures and Tables

**Figure 1 marinedrugs-19-00577-f001:**
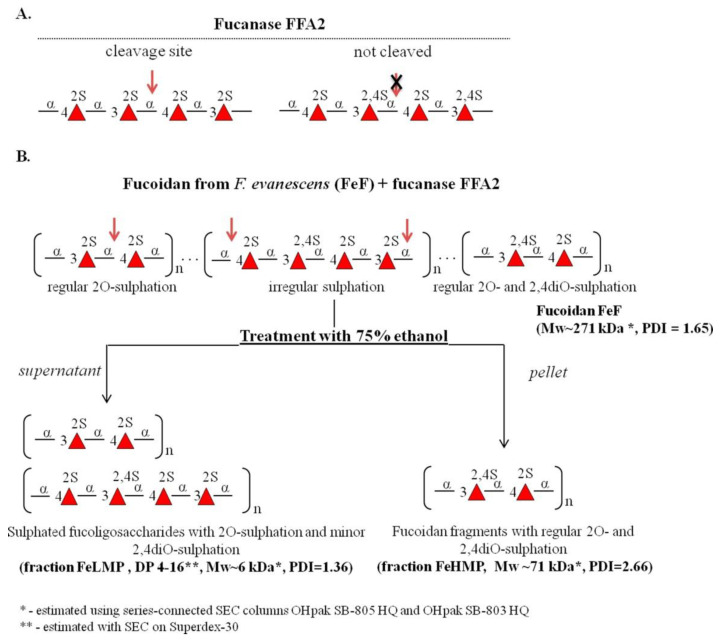
The scheme for the obtainment of the FeHMP and FeLMP fractions from fucoidan FeF using recombinant fucanase FFA2. (**A**) Specific sites in fucoidan FeF that fucanase FFA2 recognise and cleave/not cleave. (**B**) The experimental steps for the FeLMP and FeHMP fractions preparation. Red arrows indicate the proposed sites in fucoidan FeF that cleaved by fucanase FFA2. The structural characteristics of FeF and fractions FeHMP and FeLMP are indicated under the schematic representation of their structures. The Mw (average molecular weight), PDI (polydispersity index) and DP (degree of polymerisation) values are indicated under the structures of fucoidan FeF and its derivatives FeHMP and FeLMP.

**Figure 2 marinedrugs-19-00577-f002:**
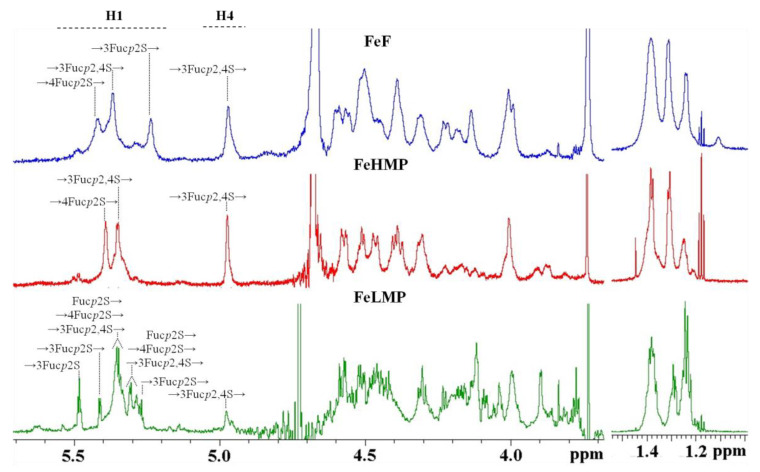
1H NMR spectra of fucoidan FeF and its enzymatically prepared derivatives FeHMP and FeLMP.

**Figure 3 marinedrugs-19-00577-f003:**
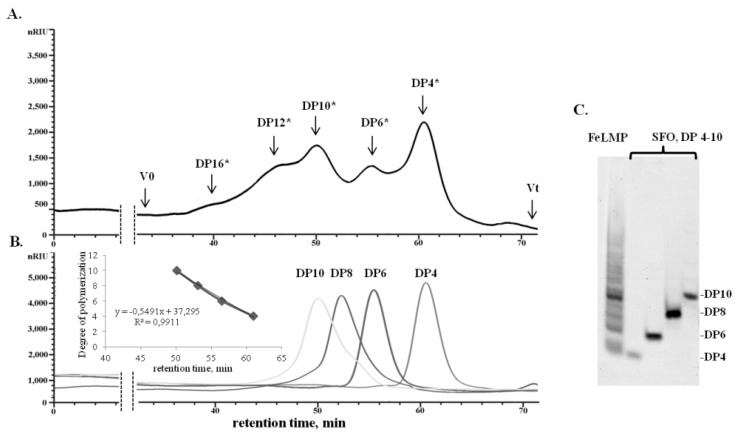
Determination of the degree of polymerisation of oligosaccharides in the FeLMP fraction. (**A**) SEC analysis of the FeLMP fraction on a chromatographic column with Superdex-30. (**B**) SEC analysis of the 2*O*-sulphated tetra-(DP = 4), hexa-(DP = 6), octa-(DP = 8) and deca-fucooligosaccharides (DP = 10) on Superdex-30. (DP *)—the calculated degree of polymerisation of oligosaccharides in the FeLMP fraction. The DP was calculated using 2*O*-sulphated fucooligosaccharides with DP from 4 to 10 as reference standard. (**C**) C-PAGE analysis of the fraction FeLMP. 2*O*-Sulphated fucooligosaccharides (SFO) with DP from 4 to 10 was used as reference standards.

**Figure 4 marinedrugs-19-00577-f004:**
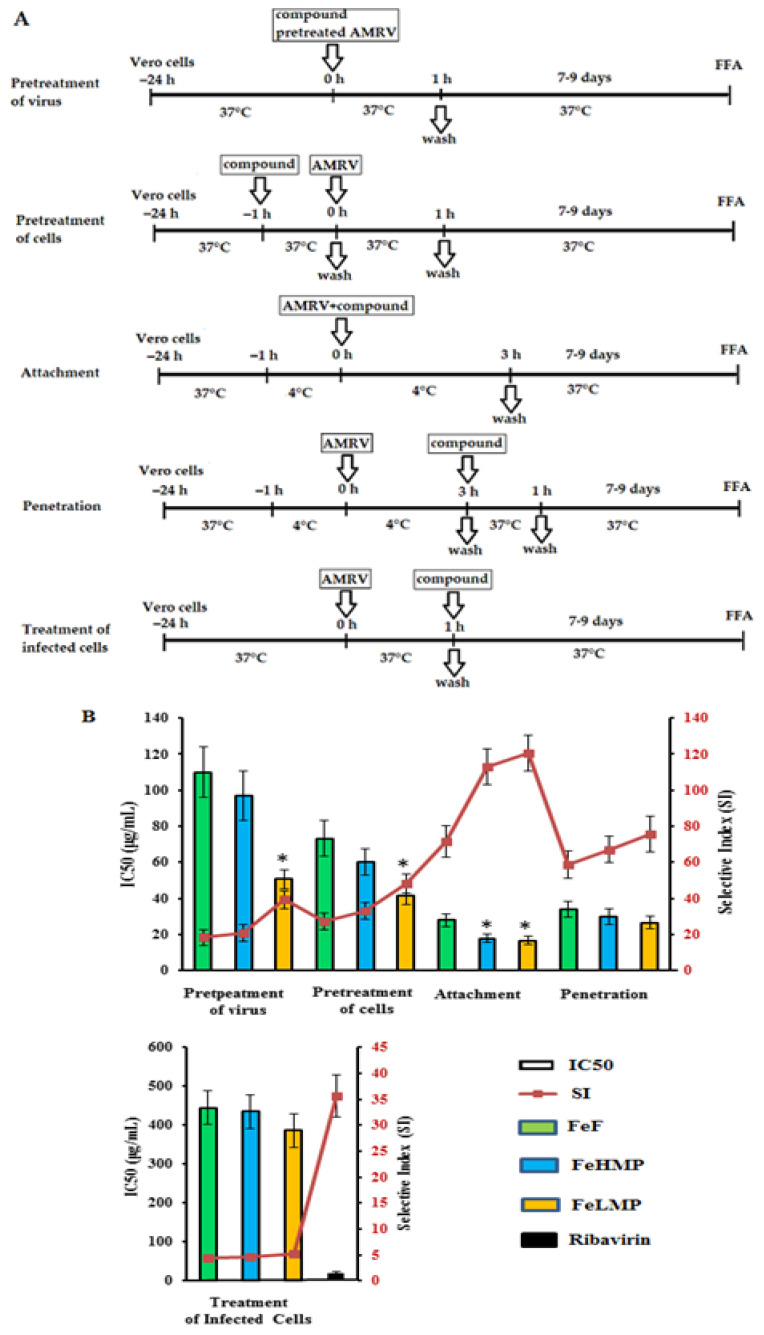
The antiviral activity of the fucoidans against the Amur virus (AMRV) at the different treatment schemes. (**A**) Vero cell monolayers were inoculated with AMRV and fucoidans at the indicated time periods and temperatures. (**B**) The anti-AMRV activity of the compounds was determined by the Focus Formation Assay (FFA) and was expressed as a 50% inhibitory concentration-IC_50_ (columns) and a selective index–SI (line). FeF, native fucoidan from brown alga *F. evanescens*; FeHMP, high molecular weight fraction and FeLMP, low molecular weight fractions of fucoidan. The results were obtained by applying fucoidans at different stages of the AMRV infection and each data point represents the mean value ± SD of three independent experiments. * Significance of the differences between the parameters of modified fucoidans (FeHMP and FeLMP) compared to native fucoidan (FeF) (*p* ≤ 0.05).

**Figure 5 marinedrugs-19-00577-f005:**
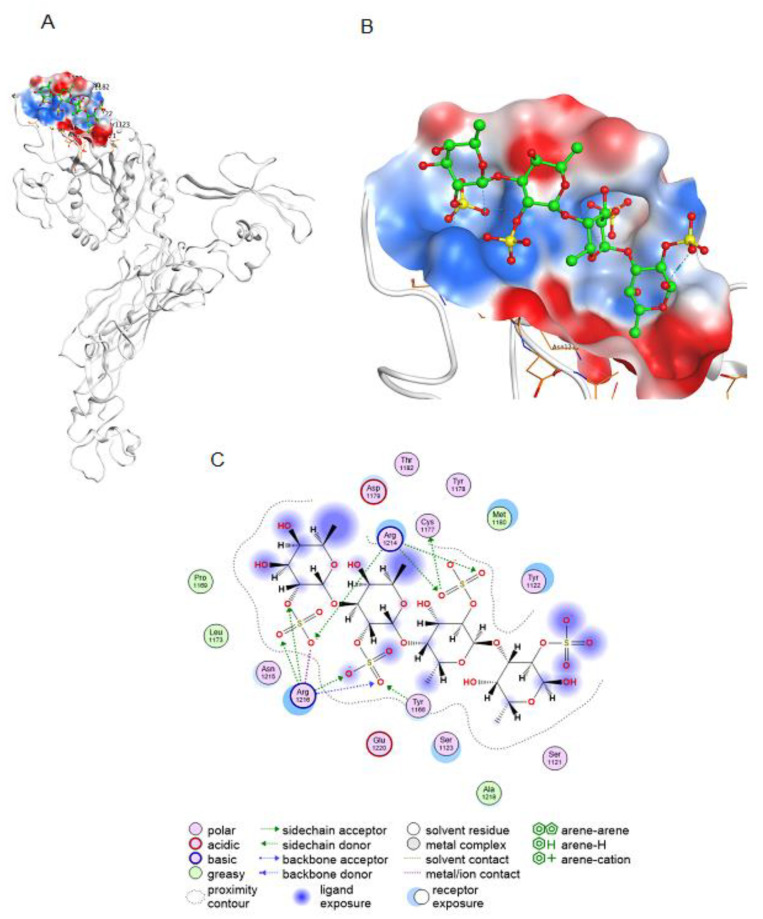
Interaction of the 2O-sulphated fucooligosaccharide with the neutralising orthohantavirus antibody epitope of the integrin β3. (**A**) Docking of 2O-sulphated fucooligosaccharide onto the integrin β3 epitope for the neutralising antibody (integrin β3 residues 177–184). (**B**) Electrostatic potential of the integrin β3 antibody-binding site. Electropositive potential is shown in blue and electronegative potential is shown in red. (**C**) Two-dimensional (2D) diagram of contacts of fucooligosaccharide at the integrin β3 binding site.

**Figure 6 marinedrugs-19-00577-f006:**
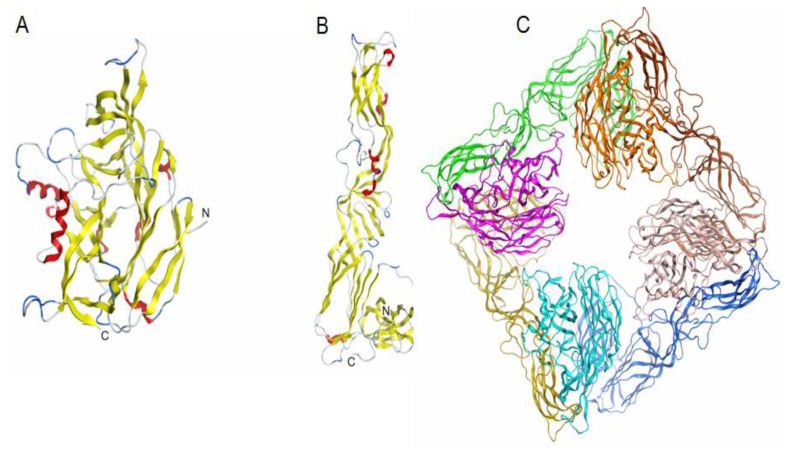
Theoretical models of the spatial structure of ectodomains of glycoproteins Gn (**A**), Gc (**B**) and a spike complex tetramer (Gn-Gc)4 (**C**) of the AMRV (Uniprot ID A3FEU7). Protein structures are shown as ribbon diagrams. The secondary structure of Gn (**A**), Gc (**B**) glycoproteins is shown in color. In the spike complex model (**C**), Gn glycoproteins are shown in orange, pink, turquoise and light pink, and Gc glycoproteins are shown in yellow, green, brown and blue.

**Figure 7 marinedrugs-19-00577-f007:**
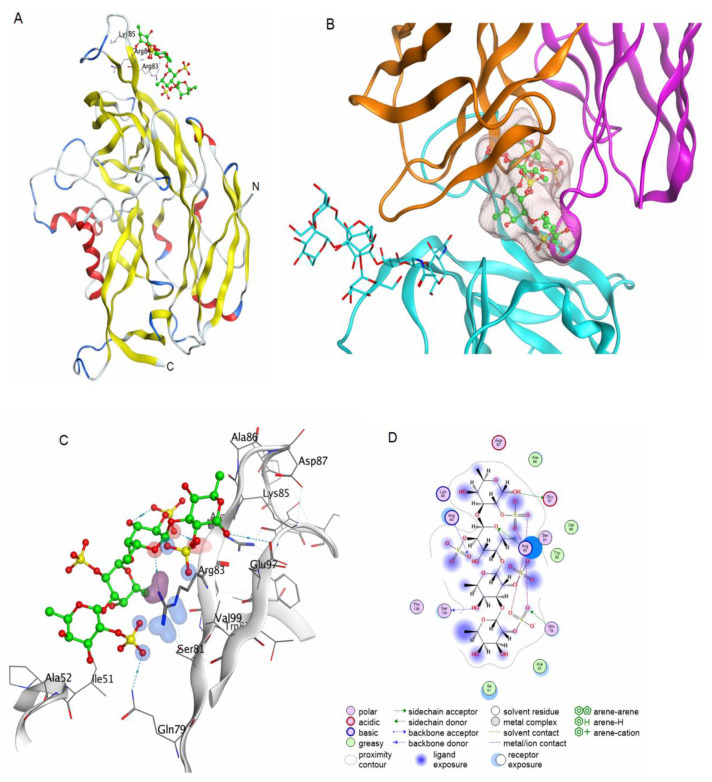
Molecular docking of a fucooligosaccharide into the AMRV Gn epitope for a neutralising antibody. Homologous model of the AMRV Gn glycoprotein and a potential fucooligosaccharide binding site (**A**). The structure of Gn is shown as ribbon, the structure of the fucooligosaccharide is shown as ball-and-stick in green. Superposition of the binding site of the fucooligosaccharide with the AMRV Gn ectodomain and the crystal structure of the complex of the orthohantavirus Gn ectodomain with a neutralising antibody (PDB ID 7nks) (**B**). The structure of the fucooligosaccharide is shown as ball-and-stick in green and the molecular surface of the oligosaccharide is shown in light pink. The 3D structure of the binding site of fucooligosaccharide and glycoprotein Gn of AMRV (**C**) and contacts of fucooligosaccharide and glycoprotein Gn (**D**).

**Figure 8 marinedrugs-19-00577-f008:**
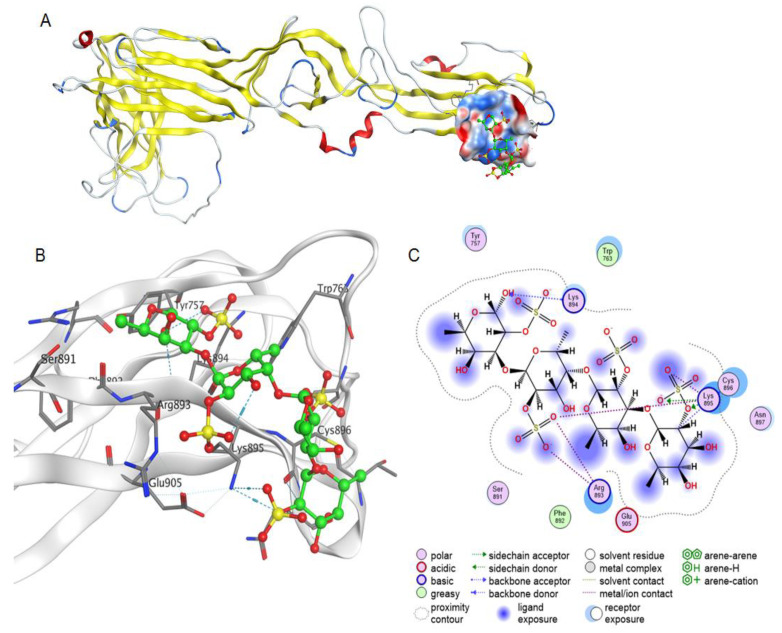
Molecular docking of a fucooligosaccharide into an epitope for a neutralizing antibody at the AMRV Gc ectodomain. Homology model of AMRV glycoprotein Gc and a potential fucooligosaccharide binding site (**A**). The Gc structure is shown as ribbon, the electrostatic potential of the molecular surface of the binding site is shown in blue and red for electropositive and electronegative sites, respectively, and the fucooligosaccharide structure is shown as ball-and-stick in green. The 3D structure of the binding site of fucooligosaccharide and AMRV Gc ectodomain (**B**) and contacts of fucooligosaccharide and glycoprotein Gc (**C**).

## Data Availability

The data presented in this study are available in this article and Appendix A.

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
