# Peer review of "In Vitro Anti-Orthohantavirus Activity of the High-and Low-Molecular-Weight Fractions of Fucoidan from the Brown Alga Fucus evanescens"

_marinedrugs, 2021, doi:10.3390/md19100577_

Round 1
Reviewer 1 Report
This work describes identification of fucoidan fractions obtained from enzymatic cleavage and their antiviral effect against orthohanta virus. While the antiviral effect is modest, potential mechanism of fucoidans as entry inhibitor was shown experimentally. Further molecular modeling work was also performed by MOE to provide a potential explanation. Antiviral effect of fucoidan against HIV or Influenza A have been demonstrated previously and this manuscript will add another virus.
While the paper is acceptable in Marine Drugs, the reviewer identified a few points to address:
- In Figure 4, what is the incubation time with the virus? The treatment scheme with timeline and arrow will give a better idea on the different experimental groups
- Role of integrin beta3 in the hantavirus entry has been demonstrated but the reviewer did not find any literature that fucoidan bind to integrin beta3. Computational model is most useful when it can explain the observed phenomena. So, target validation should be discussed in Figure 5 and the reviewer suggests in vitro biophysical binding assay of integrin beta3 and fucoidan or FACS with fluorescently labeled fucoidan using integrin b3 expressing cells. Or antiviral effect of fucoidan can be measured with integrin beta3 knock out VeroE6 cells.
- In line 216, the authors use “integrin β3 epitope ReoPro” that makes the reviewer confused. Integrin β3 epitope for ReoPro binding sounds better.
- MOE docking parameters and other docking result can be added in the supporting information. Was the structure shown in Fig 5 was the top scoring entry? How about other 4 docking poses and what is the score difference between them? The ligand interaction map was same as with other 4 docking poses? You can use superimposed ensembles of top 5 scoring structures to justify it better.
Author Response
Dear Reviewer. Thank You very much for your remarks, our comments are presented below.
- In Figure 4, what is the incubation time with the virus? The treatment scheme with timeline and arrow will give a better idea on the different experimental groups
Response:
We have presented the treatment schemes with a timeline in Figure 4.A.
2. Role of integrin beta3 in the hantavirus entry has been demonstrated but the reviewer did not find any literature that fucoidan bind to integrin beta3. Computational model is most useful when it can explain the observed phenomena. So, target validation should be discussed in Figure 5 and the reviewer suggests in vitro biophysical binding assay of integrin beta3 and fucoidan or FACS with fluorescently labeled fucoidan using integrin b3 expressing cells. Or antiviral effect of fucoidan can be measured with integrin beta3 knock out VeroE6 cells.
Response:
“Role of integrin beta3 in the hantavirus entry has been demonstrated but the reviewer did not find any literature that fucoidan bind to integrin beta3.”
It was demonstrated that Fucoidan-Sargassum inhibits metastatic behavior in the Hepatocellular carcinoma (HCC) cell lines and the mechanism underlying the Fucoidan-Sargassum-induced inhibition of invadopodia formation in HCC cells was determined to involve the deactivation of the integrin αVβ3/SRC/E2F1 signaling pathway. Fucoidan-Sargassum downregulates the Src signaling pathway by inactivating integrin αVβ3 [Pan, T. J., Li, L. X., Zhang, J. W., Yang, Z. S., Shi, D. M., Yang, Y. K., & Wu, W. Z. (2019). Antimetastatic Effect of Fucoidan-Sargassum against Liver Cancer Cell Invadopodia Formation via Targeting Integrin αVβ3 and Mediating αVβ3/Src/E2F1 Signaling. Journal of Cancer, 10(20), 4777–4792. https://doi.org/10.7150/jca.26740].
“Computational model is most useful when it can explain the observed phenomena.”
It is known that the entry of hantaviruses into human cells can be prevented by neutralizing antibodies directed against the virus [Rissanen, I.; Krumm, S.A.; Stass, R.; Whitaker, A.; Voss, J.E.; Bruce, E.A.; Rothenberger, S.; Kunz, S.; Burton, D.R.; Huiskonen, J.T.; Botten, J.W.; Bowden, T.A.; Doores, K.J. Structural Basis for a Neutralizing Antibody Response Elicited by a Recombinant Hantaan Virus Gn Immunogen. mBio 2021, 6, e0253120. doi: 10.1128/mBio.02531-20.] or against the integrin receptor αVβ3 [Gavrilovskaya, I.N.; Shepley, M.; Shaw, R.; Ginsberg, M.H.; Mackow E.R. beta3 Integrins mediate the cellular entry of hantaviruses that cause respiratory failure. Proc Natl Acad Sci USA 1998, 95(12), 7074-70799. doi: 10.1073/pnas.95.12.7074.]. We used this information when planning in silico experiments. Molecular docking of fucoidan derivatives was carried out into the sites where neutralizing antibodies interact with the virus exodomains and integrin β3.
Earlier, it was demonstrated that the antibody neutralizing hantavirus binds to integrin β3 at the ReoPro site and this leads to inhibition of hantavirus activity. It has also been shown that a small cyclic peptide that can bind to this site inhibits hantavirus activity [Larson, R.S.; Brown, D.C.; Ye, C.; Hjelle, B. Peptide antagonists that inhibit Sin Nombre virus and hantaan virus entry through the beta3-integrin receptor. J Virol. 2005, 79(12), 7319-7326. doi: 10.1128/JVI.79.12.7319-7326.2005.] [Hall, P.R.; Malone, L.; Sillerud, L.O.; Ye, C.; Hjelle, B.L.; Larson, R.S. Characterization and NMR solution structure of a novel cyclic pentapeptide inhibitor of pathogenic hantaviruses. Chem Biol Drug Des. 2007, 69(3), 180-190. doi: 10.1111/j.1747-0285.2007.00489.x.]. When cells are treated with fucoidan, inhibition of hantavirus activity is observed, so we decided to check in silico whether fucoidan binds to the integrin β3 at the site where a neutralizing antibody interacts with a peptide that inhibits hantavirus.
“…and the reviewer suggests in vitro biophysical binding assay of integrin beta3 and fucoidan or FACS with fluorescently labeled fucoidan using integrin b3 expressing cells. Or antiviral effect of fucoidan can be measured with integrin beta3 knock out VeroE6 cells.”
In the future, we will take into account the proposal of the reviewer and continue to study the interaction of fucoidan and its derivatives with cells using an in vitro biophysical binding assay and integrin beta3 knock-out Vero cells.
3. In line 216, the authors use “integrin β3 epitope ReoPro” that makes the reviewer confused. Integrin β3 epitope for ReoPro binding sounds better.
Response:
We agree with the comment of the reviewer. We changed "Integrin β3 epitope for ReoPro" instead “integrin β3 epitope ReoPro”.
4. MOE docking parameters and other docking result can be added in the supporting information. Was the structure shown in Fig 5 was the top scoring entry? How about other 4 docking poses and what is the score difference between them? The ligand interaction map was same as with other 4 docking poses? You can use superimposed ensembles of top 5 scoring structures to justify it better.
Response:
Docking parameters are given in the description of the methods: "The structures of 30 complexes with score London dG and 5 complexes with score GBVI/WSA dG were calculated".
“Was the structure shown in Fig 5 was the top scoring entry?”
Yes, the structure pose shown in Fig 5 is the top scoring.
“How about other 4 docking poses and what is the score difference between them?”
The figures below (Figures R1-1 and R1-2) shows the superposition and contact diagrams of the 2O-sulphated tetrasaccharide at the binding site.
|
Docking pose |
S |
|
1 |
"-6.0394549" |
|
2 |
"-5.856431" |
|
3 |
"-5.8465667" |
|
4 |
"-5.7499146" |
|
5 |
"-5.7247596" |
“The ligand interaction map was same as with other 4 docking poses?”
No, the ligand interaction map is different (see below Figure R1-2).
“You can use superimposed ensembles of top 5 scoring structures to justify it better.”
The Figures below shows the superposition and contact diagrams of the tetrasaccharide at the binding site.

Reviewer 2 Report
This is an informative report for preparing the FeLMP and FeHMP from FeF by using fucanase FFA2 and test of the fractions for anti-AMRV. This reviewer has some questions to be addressed by the authors.
- In Fig. 1, they showed FeP’s enzymatically products, FeLMP and FeHMP, which have defined structures. From Fig. 1B, the products at FeLMP seem larger than that at FeHMP. What are the numbers of “n” in the products of FeLMP and FeHMP? DP is degree of polymerization, but what does DPI mean?
- For pre-treatment of virus or cells with the compounds, were the compounds removed during attachment (infection of cells with the virus)? How to remove the compounds? Was it done by exchanging medium and what medium was used? Also for the “attachment” experiments, were the compounds removed after cells were infected with the virus? How long was the co-treatment? For penetration, cells were infected with the virus at 4°C and then treated with the compounds for how long? Were the compounds removed after the treatment? Please describe in more details because the interpretations depend on the detailed treatment procedure.
- In line 196-198, it states “Overall, the data obtained showed that both FeF and its enzymatically prepared derivatives, when applied during the early stages of AMRV infection, are able to inhibit viral replication effectively in Vero cells”. However, from Fig. 4, the compounds apparently inhibited virus attachment and penetration into VeroE6 cells efficiently.
- In Fig. 6, they show molecular docking of fucooligosaccharide into the epitopes for neutralising antibodies at the AMRV Gn and Gc ectodomains. This suggests that fucooligosaccharide not only binds integrin β3, but also Gn and Gc ectodomains. In Discussion, the broad-spectrum binding capability of fucooligosaccharide is due to many negatively charged sulfate oxygen atoms on the compound. Then will the compound have non-specific binding to off-targets if there is no other common structural feature except the positively charged sites of integrin β3 and ectodomains for binding with fucooligosaccharide? Please elaborate if there is a particularly structural feature for binding.
- In Fig. 6, why did they used AMRV Gn epitope for a neutralizing antibody. Is the binding site of ectodomains with integrin β3 unknown? If known, they should use the site.
- If the binding is only predicted, they may not need to show 3 Figures for modeling, but to show really experimental results, such as using BIAcore or cell-based binding assay to confirm the binding.

Author Response
Dear Reviewer. Thank You very much for your remarks, our comments are presented below and in the attachment file.
1. In Fig. 1, they showed FeF’s enzymatically products, FeLMP and FeHMP, which have defined structures. From Fig. 1B, the products at FeLMP seem larger than that at FeHMP. What are the numbers of “n” in the products of FeLMP and FeHMP? DP is degree of polymerization, but what does DPI mean?
Response:
FeHMP has a regular structure and consists of a repeating disaccharide unit, which in the scheme (Figure 1) looks more compact than the structure of FeLMP. The FeLMP structure contains more of the structural features (e.g. not regular sulphation) compare to FeHMP, so it is visually larger.
The Figure 1B showing the fragments of the structures FeF, FeHMP and FeLMP. The (n) notation has been added based on the chemical writing of formulas for the polymers. This allows understanding that this fragment can be repeated the (n)-th number of times. However, for polysaccharides, it is rather difficult to indicate the exact value of (n), since this value can vary to a large extent (for example, for FeHMP (n) is ranged from about 20 to about 180). Polysaccharides are almost always a mixture of polymers with different chain lengths. Moreover, (n) can be specified only for polymers with the regular structure, but polysaccharides are almost irregular (with the some exceptions, e.g. some O-antigens of bacterial cells etc.). For this reason, for polysaccharides more often the average molecular weight (Mw) and the polydispersity index (PDI) are indicated. This also avoids confusion, since the content of each of the chain lengths in the polysaccharide sample is not the equal.
For oligosaccharides the value of (n) is easier to define, but it’s defined for oligosaccharide with the regular chemical structure. In our case, the FeLMP fraction is a mixture of oligosaccharides with a degree of polymerization from 4 to 10; however, the structures some of them may have some minor irregularities in sulphation at C4 (lower fragment on Figure 1B for FeLMP fraction). In other words, some of the minor oligosaccharides, for an example octa- or decaccharide, can be a chimera consisting of the upper and lower fragments shown in the Figure 1B. In this case, it will be incorrect to specify the (n).
“….DP is degree of polymerization, but what does DPI mean?...”
The PDI (polydispersity index) is value shows the distribution of molecular weights in the sample of polymers. PDI can be calculated from SEC analysis (Figure S1) as Mw/Mn = PDI (where the Mw is the weight average molecular weight; Mn is the number average molecular weight). A PDI value of 1 is a “gold standard” means that polymer sample contained the polymer with fixed weight/chain length in the sample. If a PDI value is greater than 1 its means that the tested sample contains a mixture of polymers with different weights/chain length. Since polysaccharides are biosynthesized without matrices (unlike DNA or proteins), their structures and chain length in a sample can differ significantly. The PDI value for polysaccharides is always higher than 1.
We have added a description of the abbreviations Mw, PDI, and DP in the caption to Figure 1, “The Mw (average molecular weight), PDI (polydispersity index) and DP (degree of polymerization) values are indicated under the structures of fucoidan FeF and its derivatives FeHMP and FeLMP.”
2. For pre-treatment of virus or cells with the compounds, were the compounds removed during attachment (infection of cells with the virus)? How to remove the compounds? Was it done by exchanging medium and what medium was used? Also for the “attachment” experiments, were the compounds removed after cells were infected with the virus? How long was the co-treatment? For penetration, cells were infected with the virus at 4°C and then treated with the compounds for how long? Were the compounds removed after the treatment? Please describe in more details because the interpretations depend on the detailed treatment procedure.
Response:
We have included all the information you require in the Materials and Methods section 4.5.3. Anti-AMRV Activity of the Fucoidans and also in Figure 4.A.
3. In line 196-198, it states “Overall, the data obtained showed that both FeF and its enzymatically prepared derivatives, when applied during the early stages of AMRV infection, are able to inhibit viral replication effectively in Vero cells”. However, from Fig. 4, the compounds apparently inhibited virus attachment and penetration into Vero E6 cells efficiently.
Response:
We have clarified and changed the “able to inhibit viral replication effectively in Vero cells” to the “able inhibited virus attachment and penetration effectively into Vero cells”.
4. In Fig. 6, they show molecular docking of fucooligosaccharide into the epitopes for neutralising antibodies at the AMRV Gn and Gc ectodomains. This suggests that fucooligosaccharide not only binds integrin β3, but also Gn and Gc ectodomains. In Discussion, the broad-spectrum binding capability of fucooligosaccharide is due to many negatively charged sulfate oxygen atoms on the compound. Then will the compound have non-specific binding to off-targets if there is no other common structural feature except the positively charged sites of integrin β3 and ectodomains for binding with fucooligosaccharide? Please elaborate if there is a particularly structural feature for binding.
Response:
“In Discussion, the broad-spectrum binding capability of fucooligosaccharide is due to many negatively charged sulfate oxygen atoms on the compound. Then will the compound have non-specific binding to off-targets if there is no other common structural feature except the positively charged sites of integrin β3 and ectodomains for binding with fucooligosaccharide?”
The positively charged sites of integrin β3 and ectodomains for binding with fucooligosaccharide are non-specific binding sites of different hantaviruses. The results of in silico experiments showed that an increase in the number of sulphate groups in the tetrasaccharide increases the binding energy in the integrin β3 epitope. Calculated scoring is “-6.0395”, “-6.3774” and “-6.4259” for the 2О-sulphated terasaccharide (four sulphate groups), 2,4diO-sulphated tetrasaccharide (five sulphate groups) and 2,4diO-sulphated tetrasaccharide (six sulphate groups) (Attachment file, Figures R2-1, R2-2 and R2-3 below).
“Please elaborate if there is a particularly structural feature for binding.”
Molecular docking of fucoidan derivatives was carried out for the 2O-sulphated tetrasaccharide, which was the main component of the oligosaccharides mixture. Additionally, we analyzed in silico the binding ability of other oligosaccharides (minor components) with higher sulphate groups content (six and five sulphate groups per tetrasaccharide). It was found that an increase in the number of sulphate groups in the tetrasaccharide enhances the binding energy to the integrin site for RheoPro. Therefore, in the article we came to the conclusion that electrostatic interaction plays a decisive role in the binding of integrin to this site.
5. In Fig. 6, why did they used AMRV Gn epitope for a neutralizing antibody. Is the binding site of ectodomains with integrin β3 unknown? If known, they should use the site.
Response:
We used conservative Gn and Gc epitopes for many hantaviruses, for which binding of neutralizing antibodies has been shown [Li, S.; Rissanen, I.; Zeltina, A.; Hepojoki, J.; Raghwani, J.; Harlos, K.; Pybus, O.G.; Huiskonen, J.T.; Bowden, T.A. A Molecular-Level Account of the Antigenic Hantaviral Surface. Cell Rep. 2016, 15(5), 959-967. doi: 10.1016/j.celrep.2016.03.082.]. There is currently no information on the binding of specific ectodomain residues to cellular receptors responsible for viral adhesion and penetration into cells.
6. If the binding is only predicted, they may not need to show 3 Figures for modeling, but to show really experimental results, such as using BIAcore or cell-based binding assay to confirm the binding.
Response:
Antiviral activity of fucoidan and its derivatives with the established structures has been shown experimentally. Structural bioinformatics methods have predicted a high-precision 3D structure of AMRV Gn and Gc ectodomains in monomeric and tetrameric forms. Analysis of the structure of ectodomains showed that there are conservative positively charged epitopes for interaction with neutralizing antibodies. Recently, for the first time by the method of X-ray structural analysis, amino acid residues of Gn have been identified, which bind to an antibody that neutralizes hantavirus (PDB ID 7NKS). The superposition of the crystalline complex of the ectodomain with the antibody and the complex of AMRV Gn with the 2O-sulphated derivative of fucoidan obtained by molecular docking suggested that fucoidan can exhibit antiviral activity by neutralizing this epitope like an antibody.
In the future, we are planned to experimentally study the binding of ectodomains with fucoidan derivatives by various biophysical methods to verify the results obtained in silico.
We considered it possible to present computer simulation data that allow a deeper understanding at the atomic level of the interaction of AMRV Gn and Gc ectodomains and sulphated oligosaccharides exhibiting inhibitory activity against the AMRV hantavirus.

Round 2
Reviewer 2 Report
The revisions they made are acceptable.